# Stereoelectronic and hydrogen-bonding effects on hydroxyproline conformation
Fumiki Matsumura [1], Pablo Gómez Argudo [1,2], Mischa Bonn [1], Giulia Giubertoni [3] ✉ & Johannes Hunger [1] ✉

Collagen's structural integrity and biological function derive from its characteristic triple helix, governed by a unique amino acid sequence containing the rare residue (2S,4R)-hydroxyproline (4R-Hyp). Substituting 4R-Hyp with its diastereomer 4S-Hyp markedly reduces triple-helix thermal stability and impairs collagen function. The origins of this destabilization have mostly been ascribed to stereoelectronic effects such as the gauche effect and n → $\pi^*$ interactions. Here, we dissect the role of different stabilizing interactions on molecular conformation using linear infrared and two-dimensional infrared spectroscopy, aided by density functional theory calculations. We investigate the structure of N-Boc-(2S,4S)-4-hydroxyproline-methyl ester (Boc-4S-Hyp-OMe) and N-Boc-(2S,4R)-4-hydroxyproline-methyl ester (Boc-4R-Hyp-OMe) in chloroform solution, a simplified model for the water-deficient environment in collagen tissues where triple helices assemble into fibrils. We find that, while stereoelectronic effects indeed influence molecular conformation, n → $\pi^*$ interactions only moderately alter conformational equilibria. Conversely, our results show that hydrogen bonding is pivotal: the conformation of Boc-4S-Hyp-OMe is stabilized by an intramolecular hydrogen bond, whereas Boc-4R-Hyp-OMe primarily forms intermolecular hydrogen bonds, leading to a greater intermolecular affinity. These findings highlight hydrogen bonding as a key determinant of hydroxyproline conformation at the single-residue level.

Collagen is the most abundant protein in the human body, accounting for approximately 20% of total protein mass. To date, 28 different types of collagen have been identified, with Type I collagen being the most prevalent one[1]. Type I collagen has a characteristic left-handed triple helical structure[2] that consists of three individual right-handed chains as displayed in Fig. 1a. Each chain exhibits a repeating amino acid sequence in which glycine (Gly) appears at every third position (Gly-Xaa-Yaa)[3,4]. Xaa and Yaa are often 2S-Proline (Pro) or (2S,4R)-Hydroxyl proline (4R-Hyp), respectively, and Gly-Pro-4R-Hyp is the most common sequence in collagen[3]. Cells produce monomeric triple helices that are transported in the extracellular matrix, where they assemble to form higher hierarchical structures, fibrils. Fibrils then further organize into fibers that act as scaffolds of animal tissues. A proper assembly process underpins the mechanical properties of the tissues and thus their biological functionality[5].

4R-Hyp, predominantly found in collagen, is thought to play a crucial role in determining the biological functionality of collagen as it influences many of collagen's properties, including the stability of the triple helix and the self-assembly process. For instance, a deficiency or changes in the hydroxylation of Pro to 4R-Hyp is known to cause collagen diseases, such as scurvy and *Osteogenesis imperfecta*[6–8]. Notably, it has been reported that the 4R-Hyp at the Yaa position significantly enhances the thermal stability of the triple helical structure, compared to placing Pro at the same position[9,10]. However, introducing a diastereomer of the 4R-Hyp, namely (4S, 2S)-hydroxylproline (4S-Hyp), at the Yaa position (see Fig. 1b) drastically reduces the thermal stability of the triple helix. When Yaa = 4R-Hyp in (Gly-Pro-Yaa)$_{10}$, the triple helical structure is stable up to 60 ˚C, conversely, for Yaa = 4S-Hyp, the triple helix is not formed, even at room temperature[11].

To explain this drastic change in thermal stability, various explanations have been put forward: puckering of the pyrrolidine ring[12–14], hydration and hydrogen bonding[15–17], and stereoelectronic effects[18–20]. Among these mechanisms, the prevailing hypothesis is the combination of two stereoelectronic effects, namely the gauche effect and n → $\pi^*$ interactions, as summarized in Fig. 1[21]. Due to the gauche effect, placing an electronegative group at the 4 position of the pyrrolidine ring in the 4R geometry leads to the preferential $C^\gamma$- exo puckering of the ring[22,23]. In contrast, incorporating an electronegative group in the 4S geometry results in $C^\gamma$- endo puckering,

[1]Department of the Molecular Spectroscopy, Max Planck Institute for Polymer Research, Mainz, Germany. [2]Department of Physical Chemistry and Applied Thermodynamics, University of Córdoba, Córdoba, Spain. [3]Van 't Hoff Institute for Molecular Sciences, Amsterdam, The Netherlands. ✉e-mail: g.giubertoni@uva.nl; hunger@mpip-mainz.mpg.de

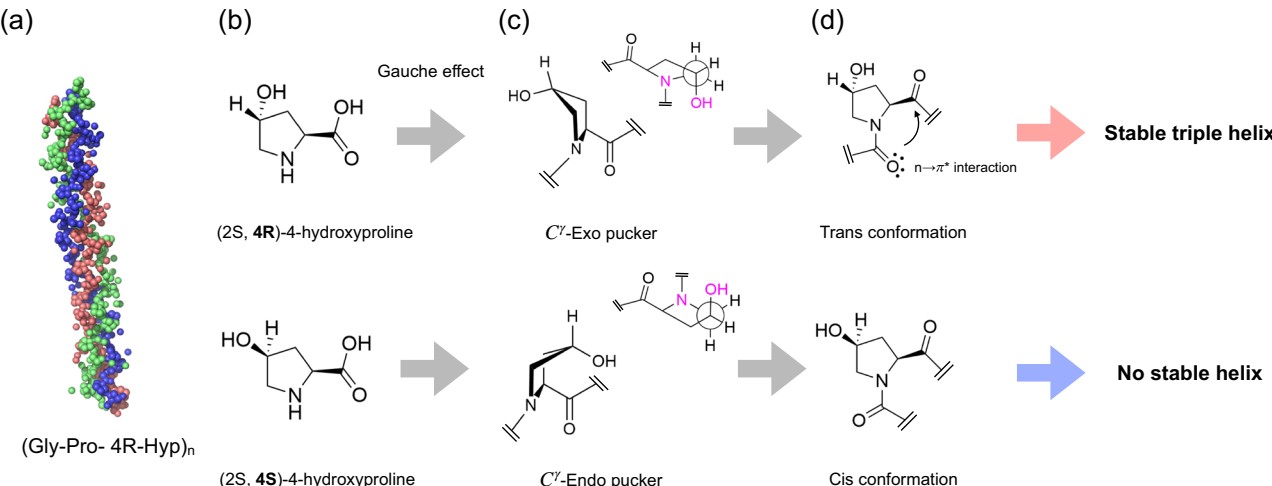

**Fig. 1 | The diastereomers of hydroxyproline and the suggested stereoelectronic mechanism of triple-helical structure. a** A characteristic amino acid sequence of collagen and a model triple helical structure rendered from the protein data bank, PDB ID: 1CAG[15]. **b** The chemical structures of the diastereomers (2S, 4R)-4-hydroxyproline and (2S, 4S)-4-hydroxyproline. **c** The structures of the gauche ($C^\gamma$-Exo pucker) and anti ($C^\gamma$-Endo pucker) conformers of the pyrrolidine ring. **d** Trans and cis conformation of the adjacent carbonyl groups. The donation of the electron lone pair from the amide oxygen ($O_{i-1}$) to the adjacent carbonyl group ($O_i$) (the proposed $n \to \pi^*$ interaction) is indicated with an arrow. The reported[11] stabilities of the triple helix of (Gly-Pro-(2S, 4R)-4-hydroxyproline)$_{10}$ and (Gly-Pro-(2S, 4S)-4-hydroxyproline)$_{10}$ at ambient temperature are indicated next to the colored arrows.

Fig. 2 | **Chemical structures of the molecules studied in this work. a** *N*-Boc- (2S, 4S)-4-hydroxyproline-methylester (Boc-4S-Hyp-OMe) and **b** *N*-Boc-(2S, 4R)-4-hydroxyproline-methylester (Boc-4R-Hyp-OMe). The dotted circles in the chemical structures are drawn to highlight the different orientations of the hydroxyl group.

(a) Boc-**4S**-Hyp-OMe

(b) Boc-**4R**-Hyp-OMe

which can be further stabilized by an intramolecular hydrogen bond[17]. These different puckers have been suggested to modulate the energetic preference of the dihedral angles of the carbonyl groups, with the $C^\gamma$- exo pucker preferring the trans conformation of the two carbonyl groups and the $C^\gamma$- endo pucker preferring the cis conformation[24]. The trans conformation can be further stabilized via $n \to \pi^*$ interactions[25]. This $n \to \pi^*$ interaction involves the donation of the electron lone pair on the carbonyl oxygen to the empty anti-bonding $\pi^*$ orbital of a nearby carbonyl group. The resulting preferences for the cis and trans conformation of 4S-Hyp and 4R-Hyp can be the reason for the different thermal stability of the (Gly-Pro-4S-Hyp)$_{10}$ and (Gly-Pro-4R-Hyp)$_{10}$. The significance of such stereoelectronic effects is demonstrated in earlier works on collagen mimetic peptides containing 4R-fluoroproline (Flp), for which fluorination enforces stereoelectronic effects without incorporating a strong hydrogen bond donor. In these studies, fluorination has been shown to markedly stabilize the triple helical structure[10,18].

In fact, these effects have been demonstrated to markedly affect the conformation of carbonyl groups in both theoretical analyses[22,26], and experiments in the gas phase[27,28]. However, the impact of such stereoelectronic effects on the structures of 4S- and 4R-Hyp, especially in solution, remains elusive. The energy of an individual $n \to \pi^*$ interaction is rather weak, and is estimated to be 0.3 kcal/mol[29]. Moreover, hydrogen bonds from water molecules to the carbonyl compete with the $n \to \pi^*$ interaction in aqueous solution[30]. Nevertheless, $^1$H-NMR spectra in deuterium oxide indicated that the trans conformer is six times more populated than the cis conformer for *N*-acetyl-4R-Hyp methyl ester, whereas the trans conformer

is only 2.4 times more populated than the cis conformer for *N*-acetyl-4S-Hyp methyl ester[19]. In contrast, in a more hydrophobic environment with weaker interactions with the solvent, such as deuterated chloroform, the trans conformer has been suggested to be approximately four times more populated than the cis conformer, for both *N*-acetyl-4R-Hyp methyl ester and *N*-acetyl-4R-Hyp methyl ester[17,31]. Based on these findings, the stereoelectronic effects and the role of interactions with the solvent on the molecular conformation cannot be rationalized in a straightforward manner. To elucidate their role in the stability of collagen, the complexity arising from conformational flexibility and solute-solvent interactions has to be disentangled also at the single-residue level.

To disentangle these effects, we examined the conformations of the different stereoisomers in detail using vibrational spectroscopies[32]. The frequency of the amide stretching peak is sensitive to the dihedral angle and the environment of the carbonyl groups, which allows us to investigate the detailed conformation of peptides[33,34]. Two-dimensional infrared (2D-IR) spectroscopy can provide additional structural information by probing the interactions between two vibrational modes[35,36]. In this study, we employed linear IR and 2D-IR spectroscopy to investigate the structures of *N*-Boc-(2S,4S)-4-hydroxyproline-methyl ester (hereafter referred to as Boc-4S-Hyp-OMe) and *N*-Boc-(2S,4R)-4-hydroxyproline-methyl ester (hereafter referred to as Boc-4R-Hyp-OMe) (Fig. 2). Measurements were performed in chloroform solution to mimic a hydrophobic environment, which does not perturb stereoelectronic effects via hydrogen bonding. To aid the interpretation of the experimental results, we used density functional theory (DFT) calculations. We found that the gauche effect indeed alters the

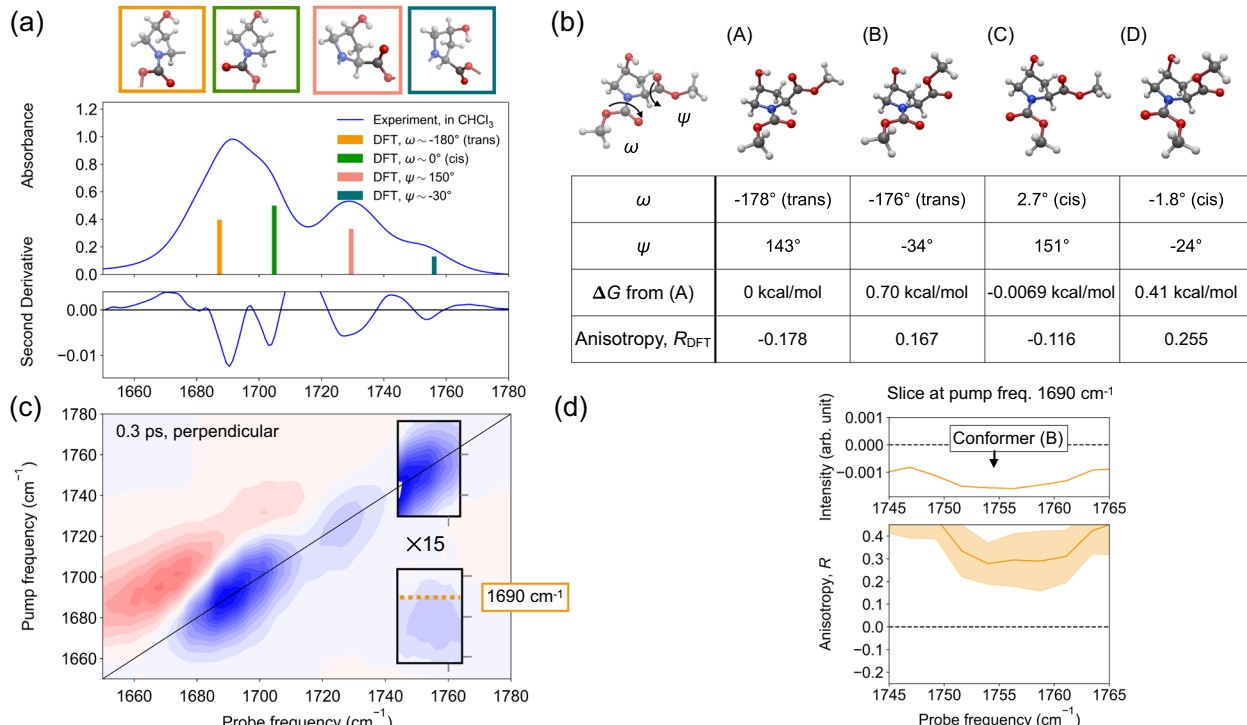

**Fig. 3 | Linear and 2D-IR spectroscopic characterization and DFT-conformer analysis of Boc-4S-Hyp-OMe in chloroform. a** Linear IR spectrum (center panel) and its second derivative (bottom panel) of a 200 mM solution of Boc-4S-Hyp-OMe in chloroform in the amide and ester stretching region. The top panel displays the conformations of the amide and ester groups, which are represented by bars of the same colors in the center panel. **b** Structures of Boc-4S-Hyp-OMe obtained from conformational search and geometry optimization by DFT calculations. In the bottom table, the dihedral angles of the amide group $\omega$ and ester group $\psi$, Gibbs free energy difference from conformer (A) $\Delta G$, and the anisotropy parameter $R_{DFT}$ calculated from the structures are displayed. Classification of conformers by the dihedral angles of the amide group $\omega$ (trans or cis) is shown in parentheses next to the dihedral angle values. **c** 2D-IR spectrum of the 200 mM Boc-4S-Hyp-OMe solution in chloroform in perpendicular polarization at a waiting time of 0.3 ps. In the gray boxes, intensities are magnified 15-fold to emphasize cross-peaks. The dotted line indicates the pump frequency corresponding to the amide stretching vibration of the trans conformer. **d** Pump slice at 1690 cm$^{-1}$ (top) and the corresponding anisotropy spectrum (bottom) as indicated by the dotted line in panel (**c**). The corresponding conformer is indicated by an arrow.

puckering of the pyrrolidine ring, and that the n → π* interaction moderately favors the trans conformer in chloroform solution. We also find the ring conformation of Boc-4S-Hyp-OMe to be stabilized by the formation of an intramolecular hydrogen bond. Conversely, Boc-4R-Hyp-OMe can only form intermolecular hydrogen bonds; this is reflected in its preference for forming intermolecular hydrogen bonds with other molecules, leading to self-aggregation for the present samples.

## Results

### Structures of Boc-4S-Hyp-OMe

To investigate the structures of Boc-4S-Hyp-OMe in chloroform, we performed linear IR measurements as displayed in Fig. 3a. The linear IR spectra show a prominent peak at 1690 cm$^{-1}$ with a shoulder at 1705 cm$^{-1}$. We also observed a distinct peak at 1730 cm$^{-1}$, with an additional shoulder at 1755 cm$^{-1}$. The negative peaks of the second derivative spectra (bottom panel of Fig. 3a) also support the presence of these peaks. To assign the peaks, we performed the conformational search of Boc-4S-Hyp-OMe, followed by geometry optimizations using DFT calculations (trajectory files optimized by DFT calculations are in Supplementary Data 2). We obtained four potential conformers in our conformational search, defined by two different dihedral angles, the angle of the amide group ($\omega \approx -180°$ or $\omega \approx 0°$) and the angle of the ester group ($\psi \approx 150°$ or $\psi \approx -30°$), as shown in Fig. 3b.

For the amide group, conformers (A) and (B) have $\omega \approx -180°$ (denoted as trans conformers following previous notations[1]), while conformers (C) and (D) are characterized by $\omega \approx 0°$ (cis conformers). The DFT calculations revealed that the trans and cis conformers have different harmonic frequencies, as indicated by the bar plots in the middle panel of Fig. 3a, with 1690 cm$^{-1}$ for the trans conformer and 1705 cm$^{-1}$ for the cis conformer. The intensities of the bar plots were computed by multiplying the squared

transition dipole moment and the population of the conformer. The populations were estimated from their relative energy (Fig. 1b), assuming a Boltzmann distribution. Based on the good agreement between the calculated frequencies and the experimental peak positions, the experimentally observed peaks were assigned accordingly to the cis and trans conformers.

Next, we investigated the dihedral angle around the ester group, $\psi$, and its consequences for the vibrational frequency. In analogy to the amide group, our DFT calculations suggest that the two distinct orientations of the ester group give rise to two distinct harmonic frequencies, as indicated by the bar plots at 1730 and 1755 cm$^{-1}$. The peak at 1730 cm$^{-1}$ is due to ester groups with $\psi \approx 150°$ (conformers (A) and (C)), and the peak at 1755 cm$^{-1}$ is due to ester groups with $\psi \approx -30°$ (conformers (B) and (D)). Based on the good agreement of the computed and experimental frequencies, the peaks in the experiment were assigned accordingly. These two frequencies are markedly separated (by $\approx 30$ cm$^{-1}$). In addition to the conformational effect, this shift upon rotation of $\psi$ can be ascribed to intramolecular hydrogen bonding between the OH group and the ester C = O[34]: For the experiments, the effect of the hydrogen bond formation to the solvent is limited, since chloroform can only act as a weak hydrogen bond donor[37,38]. Thus, hydrogen bonds can be either intra- or intermolecular hydrogen bonds with the OH group of Boc-4S-Hyp-OMe. In fact, the structures from our DFT calculations suggest that intramolecular hydrogen bond formation between the carbonyl oxygen of the ester group and the hydrogen of the hydroxyl group can explain the observed frequencies: In conformers (A) and (C), the distance between the ester carbonyl oxygen and the hydroxyl hydrogen is $\approx 2$ Å, which indicates the formation of a strong intramolecular hydrogen bond[39]. Unlike conformers (A) and (C), the ester carbonyl oxygen does not form such a hydrogen bond in conformers (B) and (D). Indeed,

such intramolecular hydrogen bond formation is consistent with the previous computational and experimental studies on similar molecules[17,19,31,40].

Our analysis of the linear IR spectrum revealed that Boc-4S-Hyp-OMe shows distinct peaks of amide and ester groups, and the frequencies of the peaks reflect their dihedral angles and intramolecular hydrogen bond formation. However, linear IR spectroscopy does not provide information about the relative orientation of the vibrational modes, required to fully elucidate the molecular conformation. To elucidate these orientations, we use polarization-resolved 2D-IR spectroscopy. Figure 3c shows the 2D-IR spectrum of Boc-4S-Hyp-OMe with perpendicular probe polarization at a waiting time of 0.3 ps. In Fig. 3c, the blue contours indicate bleaching signals, whereas the red contours show induced absorptions. At the diagonal of the spectrum, we observed an intense bleaching signal centered at 1690 cm$^{-1}$ adjacent to a weak bleaching signal at 1725 cm$^{-1}$. Furthermore, we detected a weak bleaching signal at 1750 cm$^{-1}$, as highlighted in the box in the plot. These bleaching signals (due to ground-state bleach and stimulated emission) are accompanied by excited-state absorption peaks at lower probe frequencies, red-shifted due to anharmonic shifts[41]. The frequencies of the diagonal signals are consistent with the peak positions inferred from the linear IR spectra, and were assigned accordingly.

In the off-diagonal region of the 2D-IR spectra, we also observed bleaching signatures (highlighted by the box in Fig. 3c), which we refer to as cross-peaks hereafter. We observed a cross-peak in the spectra for excitation at 1690 cm$^{-1}$ and detection at 1755 cm$^{-1}$, indicative of coupling between the vibrations at these frequencies. We note that we also detected a finite signal at a pump frequency of 1690 cm$^{-1}$ and a probe frequency of 1725 cm$^{-1}$; however, closer inspection of these spectral contributions shows that they are dominated by the tail of the diagonal signals. As such, these signatures cannot be ascribed solely to coupling, and we therefore did not analyze these spectral contributions in detail. In addition to the contribution of the tail of the diagonal peaks to the cross-peaks, also overlapping spectral contributions in the diagonal peaks prevent a more quantitative evaluation of the coupling strengths between amide and ester vibrational modes in our system.

We then calculated the anisotropy of the cross-peaks, defined as $R = \frac{I_{\text{par}} - I_{\text{perp}}}{I_{\text{par}} + 2I_{\text{perp}}}$, where $I_{\text{par}}$ and $I_{\text{perp}}$ are the 2D-IR signal intensities recorded in parallel and perpendicular polarizations, respectively (2D-IR parallel spectra are shown in Supplementary Figs. 5, 6, and Supplementary Note 4). In the case of cross-peaks, the anisotropy can be used to estimate the angle, $\theta$, between the transition dipoles of the two coupled vibrations as $R = (3\cos^2\theta - 1)/5$[42–44]. To obtain $R$ for the cross-peak observed when exciting at 1690 cm$^{-1}$ and probing at 1755 cm$^{-1}$, we took a slice of the 2D-IR spectrum at a pump frequency of 1690 cm$^{-1}$ (top panel of Fig. 3d). We chose this pump frequency based on the peak position in the second derivative of the linear IR spectra. The calculated anisotropy spectrum $R$ for this slice (Fig. 3d) indicates an anisotropy value of $\approx 0.3$ around 1755 cm$^{-1}$. Note that the shaded area in the plot indicates the experimental standard deviation. According to the frequency of the cross-peak, the peak should arise from conformer (B). To further validate this assignment, we compared the experimental anisotropy $R$ with the anisotropy computed from our DFT calculations $R_{\text{DFT}}$. The anisotropy calculated for conformers (A) and (D) (Fig. 3b) clearly deviate from the experimental value at 1755 cm$^{-1}$ of $R \approx 0.3$. However, although broadly consistent within experimental uncertainty, the calculated value for conformer (B) of $R_{\text{DFT}} = 0.167$ is somewhat lower than the experimental value. In fact, the experimental value is closer to conformer (D). Yet, we note that the calculated anisotropy $R_{\text{DFT}}$ should also be considered as an estimate, as it is based on an isolated molecule embedded in a continuum solvent. In fact, as we show in the Supplementary Information (Supplementary Table 1 and Supplementary Note 1), accounting for explicit solvent molecules can increase the anisotropy by $\approx 0.1$, resulting in good agreement between the experimental value and the calculated value for the anticipated conformer (B). As such, the 2D-IR experiments are in line with conformational analysis of Boc-4S-Hyp-OMe based on the vibrational frequencies.

## Structures of Boc-4R-Hyp-OMe

In analogy to our results for Boc-4S-Hyp-OMe, we investigated the structure of Boc-4R-Hyp-OMe in chloroform solution. The IR spectrum of Boc-4R-Hyp-OMe in Fig. 4a is similar to that of Boc-4S-Hyp-OMe. Hence, for brevity, we only focus on the differences: The spectrum of Boc-4R-Hyp-OMe shows an additional shoulder at 1675 cm$^{-1}$, most apparent in the second derivative spectrum. Conversely, the peak at 1725 cm$^{-1}$ is largely absent. To perform spectral assignments, we again employed DFT calculations, which are summarized in Fig. 4b, and the resulting harmonic frequencies are shown as bar plots in Fig. 4a (trajectory files optimized by DFT calculations are in Supplementary Data 2).

Our DFT calculations successfully reproduced the peak positions of the peaks at 1690, 1705, and 1750 cm$^{-1}$, while the peak at 1675 cm$^{-1}$ was not found. The peaks at 1690 cm$^{-1}$ and 1705 cm$^{-1}$ are assigned to the amide stretching mode of the trans (conformers (E) and (F)) and cis (conformers (G) and (H)) conformers, respectively. The 1750 cm$^{-1}$ peak was assigned to the ester stretching vibration, which can adopt two conformations with different $\psi$. The lower frequency side is dominated by ester groups having $\psi \approx -30°$ (conformers (F) and (H)), and the higher frequency side arises from the ester group having $\psi \approx -155°$ (conformers (E) and (G)). The separation of these two components is only around 2.5 cm$^{-1}$, much smaller than in Fig. 3a, consistent with the absence of intramolecular hydrogen bonds.

The 2D-IR spectrum of Boc-4R-Hyp-OMe in the perpendicular polarization at a waiting time of 0.3 ps (Fig. 4c) again shows the bleaching signals at 1690 cm$^{-1}$ and 1750 cm$^{-1}$ along the diagonal, with the corresponding red-shifted excited-state absorptions, consistent with the linear IR spectrum. In addition to these diagonal peaks, we found cross-peak signals centered at pump frequencies of 1690–1705 cm$^{-1}$ and a probe frequency of $\sim 1750$ cm$^{-1}$. Analogous to the cross-peak of Boc-4S-Hyp-OMe, these signals can arise from vibrational coupling of the amide and ester stretching modes. To identify the underlying conformers, we investigated the slices of the 2D-IR spectrum at pump frequencies of 1690 cm$^{-1}$ and 1705 cm$^{-1}$, corresponding to the amide frequency of the trans and cis conformers, respectively. The slice at 1690 cm$^{-1}$, together with its anisotropy $R$ (Fig. 4d), indicates a value of $R \approx 0.20$. Given the amide frequency of 1690 cm$^{-1}$, the cross-peak can stem from conformer (E) or (F). DFT calculations predicted an anisotropy $R_{\text{DFT}}$ of -0.18 for conformer (E) and 0.12 for conformer (F), suggesting that conformer (F) prevails. The pump slice at 1705 cm$^{-1}$, the frequency of the amide mode of the cis conformer, is displayed together with its anisotropy in Fig. 4e. The DFT calculations suggest that this signal can be due to conformer (G) or (H). Experimentally, we obtained $R \approx 0.3$ as indicated in the bottom panel of Fig. 4e, while DFT calculations suggest anisotropies $R_{\text{DFT}}$ of –0.0044 for conformer (G) and 0.24 for conformer (H). As such, we excluded conformer (G), since the experimental and computational anisotropies did not match, even taking experimental uncertainties into account. Conversely, the cross-peak pumped at 1705 cm$^{-1}$ can be ascribed to conformer (H).

Our analysis of the 2D-IR spectrum suggests the cross-peak mainly arises from conformers (F) and (H). This may be somewhat surprising, as conformer (F) has the highest Gibbs free energy of the four conformers. As such, our experiments suggest that the calculated Gibbs free energies do not accurately predict the populations in solution. This discrepancy is further underlined by the vibrational spectra: While the DFT-calculated frequencies agree well with the experimentally inferred peak positions of the 1690 cm$^{-1}$ (trans) and 1705 cm$^{-1}$ (cis) peaks, the predicted intensities from the DFT calculations do not match the experimentally observed intensities. The reason for the inaccuracy of the DFT energy is most likely related to shortcomings of the continuum solvent approach. Hence, experiments are indispensable to elucidate conformational equilibria. In addition to the discrepancies in the populations, the DFT calculations did not reproduce the shoulder at 1675 cm$^{-1}$ observed in the experiment. As this red-shifted amide

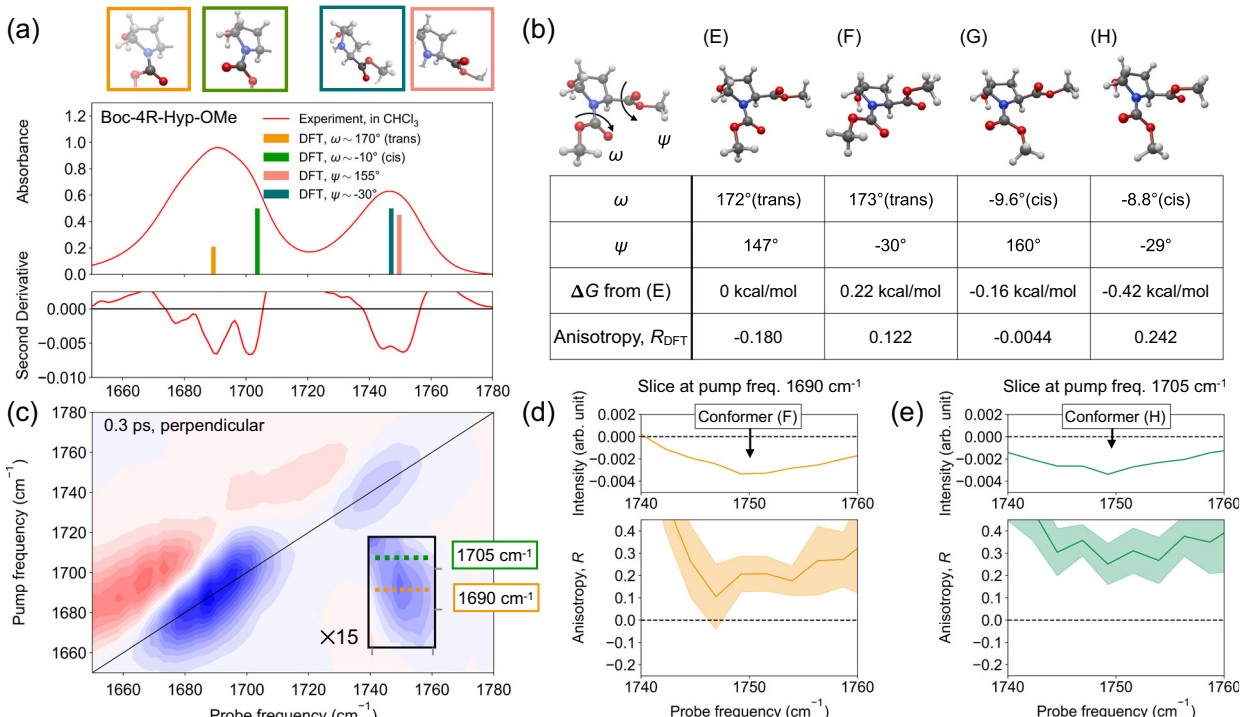

**Fig. 4 | Linear and 2D-IR spectroscopic characterization and DFT-conformer analysis of Boc-4R-Hyp-OMe in chloroform. a** Linear IR spectrum (center panel) and the second derivative (bottom panel) of 200 mM Boc-4R-Hyp-OMe in chloroform in the amide and ester stretching region. The top panel shows the conformations of the amide and ester groups underlying the bar plots of the same color in the center panel. **b** The structures of Boc-4R-Hyp-OMe obtained from conformational search and geometry optimization by DFT calculations. In the bottom table, the dihedral angles of the amide group $\omega$ and ester group $\psi$, Gibbs free energy difference from conformer (E) $\Delta G$, and the anisotropy parameter $R_{DFT}$ calculated from the structures are displayed. Classification of conformers by the dihedral angles of the amide group $\omega$ (trans or cis) is shown in parentheses next to the dihedral angle values. **c** 2D-IR spectrum of 200 mM Boc-4R-Hyp-OMe in chloroform in perpendicular polarization at a waiting time of 0.3 ps. In the gray boxes, intensities are magnified 15-fold to emphasize cross-peaks. The dotted lines indicate the pump frequency corresponding to the amide stretching vibration of the trans (orange) and cis (green) conformers. Pump slices (top) and anisotropy spectra (bottom) at (**d**) 1690 cm$^{-1}$ and **e** 1705 cm$^{-1}$, as indicated by the dotted lines in panel (**c**). The corresponding conformer for the peak is indicated with an arrow.

frequency is indicative of strong interactions with its environment, we hypothesized that it arises from intermolecular interactions between the amide group of a Boc-4R-Hyp-OMe molecule and the hydroxyl group of another molecule (self-aggregation), which cannot be captured by the single-molecule DFT calculations.

### Different aggregation tendencies of Boc-4S- and Boc-4R-Hyp-OMe

To test this hypothesis, we examined the concentration dependence of the amide stretching bands of Boc-4S- and Boc-4R-Hyp-OMe as presented in Fig. 5a, b. The concentration-dependent spectra of Boc-4S-Hyp-OMe exhibited only minor changes in spectral shape: the intensity at the higher frequency side of the peak around 1705 cm$^{-1}$ decreased, and the intensity at the lower frequency side of the peak around 1675 cm$^{-1}$ increased with increasing concentration. In contrast, for Boc-4R-Hyp-OMe, we observed a more pronounced decrease in the intensity around 1705 cm$^{-1}$ and an increase around 1675 cm$^{-1}$ with increasing concentration. Based on the concentration-dependent increase of the intensity of the peak at 1675 cm$^{-1}$ of Boc-4R-Hyp-OMe, we assigned this peak to the amide group forming intermolecular hydrogen bonds.

To confirm the donation of hydrogen bonds of the hydroxyl groups, we investigated the concentration dependence of the IR spectra of Boc-4S- and Boc-4R-Hyp-OMe in the OH stretching region as presented in panels (c) and (d) in Fig. 5. Both Boc-4S- and Boc-4R-Hyp-OMe showed two distinct peaks: a broad peak centered at 3450 cm$^{-1}$ and a narrow peak at 3600 cm$^{-1}$. At 10 mM concentration, the peak at 3450 cm$^{-1}$ prevails for Boc-4S-Hyp-OMe, and only a weak contribution at 3600 cm$^{-1}$ is discernible. Conversely, at 10 mM Boc-4R-Hyp-OMe, the 3600 cm$^{-1}$ peak of Boc-4R-Hyp-OMe is more pronounced than the 3450 cm$^{-1}$ peak. Remarkably, Boc-4R-Hyp-

OMe showed a significant concentration-dependent change in the lineshape: the intensity of the 3450 cm$^{-1}$ peak increases with increasing concentration, while the 3600 cm$^{-1}$ peak decreases. Conversely, the spectra of Boc-4S-Hyp-OMe are largely unaffected by altered concentration. Based on their frequencies, the broad peak at 3450 cm$^{-1}$ is assigned to hydroxyl groups engaged in hydrogen bonding, whereas the sharp peak at 3600 cm$^{-1}$ is assigned to free hydroxyl groups that do not participate in hydrogen bonding[45]. As such, the shift of the intensity from 3600 cm$^{-1}$ to 3450 cm$^{-1}$ with increasing concentration for Boc-4R-Hyp-OMe reflects an increasing population of hydroxyl groups engaged in hydrogen bonding. From the concentration dependence, we conclude that these hydrogen bonds are intermolecular. Hence, we find evidence for Boc-4R-Hyp-OMe having a marked aggregation tendency. In contrast, the hydrogen-bonded OH band at 3450 cm$^{-1}$ dominates at all concentrations for Boc-4S-Hyp-OMe, in line with the formation of an intramolecular hydrogen bond.

### Discussion

Firstly, we discuss the puckering of the pyrrolidine ring of Boc-4S- and Boc-4R-Hyp-OMe. In literature, it has been suggested that the gauche effect leads to the $C^\gamma$-endo pucker when an electronegative group is placed at the 4 carbon of the pyrrolidine ring in the 4S geometry, while it leads to the $C^\gamma$-exo pucker if an electronegative group is placed in the 4R geometry[22,23]. The structures obtained from our DFT calculations indeed suggest that the pyrrolidine ring prefers $C^\gamma$-exo puckering for Boc-4R-Hyp-OMe, while for Boc-4S-Hyp-OMe $C^\gamma$-endo puckering is favored, in line with the previous reports[12–14]. Notably, based on our experimental and computational results, Boc-4S-Hyp-OMe shows an intramolecular hydrogen bond between the hydroxyl group and the oxygen of the ester carbonyl group. This intramolecular hydrogen bond further stabilizes the $C^\gamma$-endo pucker of Boc-4S-Hyp-OMe, in line with what

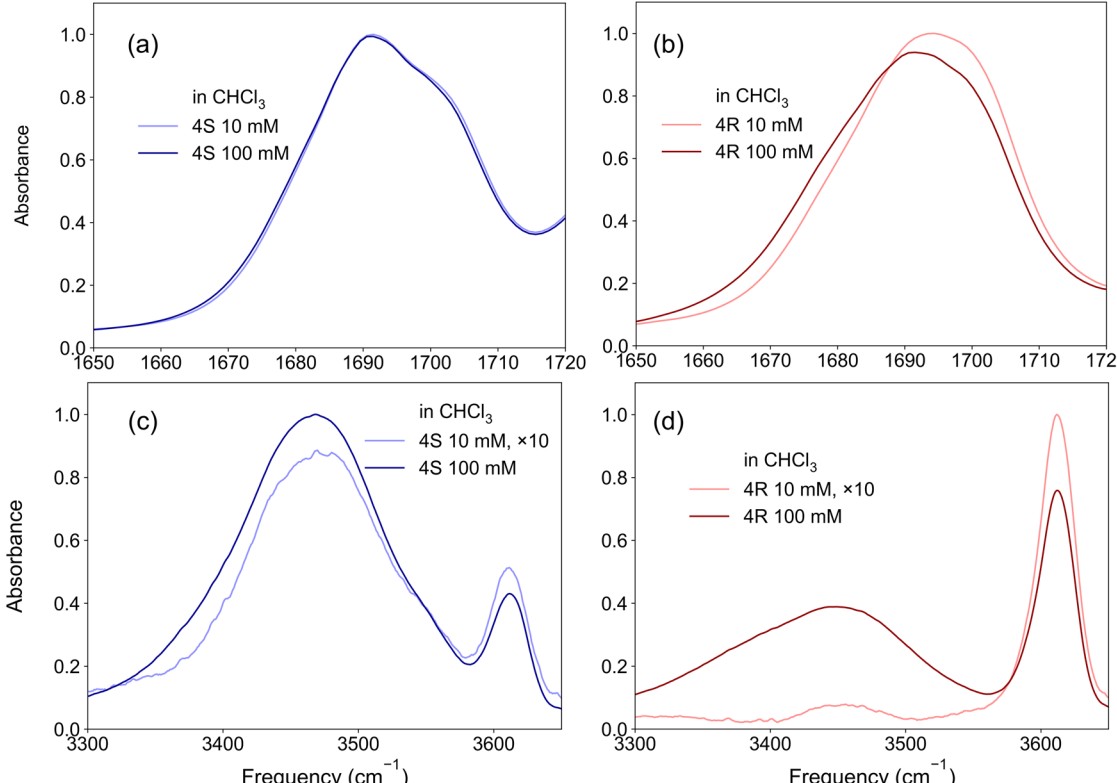

**Fig. 5 | Concentration-dependent linear IR spectra of Boc-4S- and Boc-4R-Hyp-OMe in chloroform.** The concentration-dependent (10 and 100 mM) linear IR spectra of Boc-4S- and Boc-4R-Hyp-OMe in chloroform solution in the amide stretching (**a**) and **b** and OH stretching (**c**) and **d** region. The IR spectra were normalized to the total area of the peaks in the displayed region for panels (**a**) and (**b**). Due to the largely differing transition dipole moments of hydrogen-bonded and non-hydrogen-bonded OH stretching modes, the spectra in panels (**c**) and (**d**) were scaled to account for the different concentrations.

has been found computationally[17,40] and supported by experiments[17,19,31]. Our experimental results suggest that such hydrogen bonding also markedly affects the conformation of the ester group, as defined by $\psi$. In the linear IR spectra (Fig. 3a), the peak of the hydrogen-bonded ester group at 1725 cm$^{-1}$ is much more intense than the peak at 1755 cm$^{-1}$, indicating the predominant population of the ester group of $\psi \approx 150°$, with an intramolecular hydrogen bond. Conversely, our results do not reveal a preference for the conformation of the ester in the case of Boc-4R-Hyp-OMe. As such, we find that the intramolecular hydrogen bond not only determines the ring geometry but also the orientation of the ester group. We highlight the importance of an intramolecular hydrogen bond of Boc-4S-Hyp-OMe.

Next, we examine the dihedral angle of the amide group $\omega$, which defines the trans or cis conformers. In contrast to the cis conformer, the trans conformer has been proposed to be stabilized via n $\rightarrow$ $\pi^*$ interactions by donating the electron lone pair of the carbonyl oxygen to the empty $\pi^*$ orbital of another carbonyl group[21]. If this stabilization were significant, we would expect a pronounced population of the trans conformer for both Boc-4S- and Boc-4R-Hyp-OMe. Our results demonstrate that these conformers appear at different frequencies in the vibrational spectra, as presented in Figs. 3a and 4a. To evaluate the population of trans and cis conformers, we performed peak fitting analysis. The area of each peak component from the peak fitting should be proportional to the population of the conformer, since our DFT calculations suggest that the transition dipole moments of the amide modes are comparable for trans and cis conformers (see Supplementary Table 2 and Supplementary Note 2). Although peak fitting is a powerful approach for evaluating the populations of spectral components, we found that the amplitudes somewhat depend on the assumed lineshapes. Thus, we conducted peak fitting using different assumptions for the lineshapes as summarized in the Supplementary Information (Supplementary Figs. 1–4 and Supplementary Note 3). Our results show that the ratios of trans and cis conformers range from 1.1 to 2.4 for Boc-4S-Hyp-OMe and

from 1.4 to 1.6 for Boc-4R-Hyp-OMe, depending on the assumed lineshape. The excess of trans conformers suggests that n $\rightarrow$ $\pi^*$ interactions indeed effectively stabilize the trans conformer for both Boc-4S- or Boc-4R-Hyp-OMe. In fact, our results are in broad agreement with the notion that an individual n $\rightarrow$ $\pi^*$ interaction between carbonyl groups is relatively weak ($\approx 0.3$ kcal/mol)[29], which would correspond to a trans/cis ratio of ~ 1.6. It is also worth noting that our results challenge conclusions from a previous $^1$H-NMR study of two considered conformers in deuterated chloroform, which have suggested that there is approximately 4 times more population of the trans conformer than the cis conformer, regardless of N-acetyl-4S- or N-acetyl-4R-Hyp methyl ester[17,31]. We speculate that these differences stem in part from slight differences in the investigated molecules, which can give rise to altered intermolecular interactions.

Finally, although the single-residue level of the present work cannot be directly related to larger systems such as full collagen fibrils, our results may have potential implications for larger systems. Our results indicate that Boc-4R-Hyp-OMe tends to form intermolecular interactions resulting in enhanced aggregation as compared to Boc-4S-Hyp-OMe. In full collagen fibrils, the conformational constraints due to other residues and proteins render such direct interactions between collagen residues very unlikely. Nevertheless, the self-aggregation observed here reflects a pronounced tendency of Boc-4R-Hyp-OMe to expose its hydroxyl group to the environment, resulting in intermolecular interactions rather than intramolecular interactions. This observation is consistent with the notion that naturally occurring 4R-Hyp in collagen functions as a keystone within the hydration network and as a potential recognition site during assembly[15,46], both of which require exposure of 4R-Hyp's OH group to the environment.

## Conclusion
We investigated the structure of Boc-4S- and Boc-4R-Hyp-OMe molecules in chloroform solutions to understand the role of stereoelectronic effects on

**Article**

the conformation in a water-deficient environment. Experiments together with conformational searches combined with DFT calculations suggest the tendency for $C^\gamma$- endo and -exo puckering of Boc-4S- and Boc-4R-Hyp-OMe, respectively. Furthermore, we found that weak $n \rightarrow \pi^*$ interactions moderately stabilize the trans conformations in solution for both Boc-4S- and Boc-4R-Hyp-OMe. In addition to these stereoelectronic effects, our results provide direct experimental evidence for the different ring conformations being influenced by intramolecular hydrogen-bonding: The hydroxyl group of Boc-4S-Hyp-OMe can form an intramolecular hydrogen bond, whereas that of Boc-4R-Hyp-OMe can only form intermolecular hydrogen bonds. These differences in hydrogen bonding result in markedly different intermolecular affinity in solution of the two molecules. Thus, our findings not only refine our understanding of the stabilizing mechanisms of the hydroxyproline conformation but highlight the importance of hydrogen bonding in determining its conformational equilibria and molecular interaction tendencies at the level of a single amino acid residue.

## Methods

### Sample preparation
N-Boc-(2S, 4R)-4-hydroxyproline-methylester ( > 97%, Tokyo Chemical Industry Co., Ltd.) and N-Boc-(2S, 4S)-4-hydroxyproline-methylester ( > 95.0%, Tokyo Chemical Industry Co., Ltd.) were used without further purification. Chloroform ( > 99.8%, Fischer Scientific) was dried with molecular sieves (4 Å, Carl Roth GmbH). Molecular sieves were dried in an oven at 250 °C for 2 h and placed in the solvents overnight. Samples with concentrations of 10, 100, and 200 mM were prepared by weight, assuming that the volume of the solute molecule is negligible for dilute solutions.

### Linear infrared (IR) spectroscopy
Linear infrared absorption (IR) spectra were measured using a Bruker Vertex 70 spectrometer in transmission geometry. The spectra were recorded with a resolution of $4 \text{ cm}^{-1}$ at frequencies ranging from 400 to $4000 \text{ cm}^{-1}$. The sample compartment was purged with dried air during the measurement. Samples were contained between two $CaF_2$ windows, and the thickness of the sample was controlled to 50 μm for the 100 and 200 mM solutions, and 200 μm for the 10 mM solution, using a Teflon spacer.

### Two-dimensional infrared (2D-IR) spectroscopy
The two-dimensional infrared setup is based on 800 nm laser pulses (7 W, 35 fs, 1 kHz) from a regenerative amplified laser system (Coherent, Astrella). 2.7 W of the 800 nm pulses were used to pump an optical parametric amplifier (TOPAS Prime, Coherent), to generate signal and idler pulses. Signal and idler pulses were used to generate infrared pulses at ~ 6000 nm (18 μJ, $400 \text{ cm}^{-1}$ FWHM) using non-collinear difference frequency generation (NDFG Topas, Coherent). The IR beams are guided into a commercial 2D infrared spectrometer (2D Quick IR, Phasetech Inc.).

In the 2D-IR spectrometer, the reflection from a wedged ZnSe window is used as a probe beam. The transmitted IR light is guided to a pulse shaper, where it is diffracted from a grating (150 l/mm), collimated using a parabolic mirror, and guided to a Germanium-based acousto-optic modulator (AOM). The IR light is diffracted from the AOM and focused onto a second grating (150 l/mm). The shaped beam is reflected from a retroreflector on a translational stage to control the waiting time ($T_2$) between the pump and the probe beams. After setting the polarization of the pump beam to 45° relative to the probe beam polarization using a $\lambda/2$ plate, the pump and the probe beams are focused into the sample using an off-axis parabolic mirror. After the sample, the parallel and perpendicular components are collected sequentially by rotating a polarizer in the probe beam path by 90°. Each probe component is focused onto the slit of an imaging spectrograph (SP2156 spectrograph, Princeton Instruments, 30 l/mm grating) and detected using a liquid-nitrogen-cooled 128 × 128 pixel mercury cadmium telluride array detector.

Frequency resolution on the pump axis is obtained in the time domain, using the pulse shaper to generate two pump pulse pairs delayed by 0 to 1925

fs at increments of 35 fs. To reduce data acquisition time, we used a rotating frame at $1450 \text{ cm}^{-1}$. Pump pulses were corrected for group-velocity dispersion, and the dispersion parameters were optimized using the transient signal generated from multi-photon absorption in a 0.5 mm-thick Ge plate. Before transforming the free induction decays into the frequency domain, the time-domain data were zero-padded to 128 data points and filtered with a Hamming window. The resulting intensity of the 2D-IR signals is proportional to the difference between the probe absorption with and without the pump pulse.

### Density functional theory calculation
DFT calculations were performed by the ORCA software package[47]. To explore conformational degrees of freedom of the molecules, we used the CREST software package[48] with the GFN2-xTB method[49]. Geometry optimizations of the obtained structures from the conformational search were done with the M06-2x functional[50] and the cc-pVTZ basis set[51]. We used the conductor-like polarizable continuum model (CPCM)[52] to include the effect of solvation. Harmonic vibrational frequencies from DFT calculations were scaled by a factor of 0.9589[30].

## Data availability
All raw data generated and analyzed throughout this work are available from the corresponding authors upon reasonable request. The source data for Figs. 3, 4, and 5 are provided as Supplementary Data 1. Coordinate files of the DFT optimized geometries are provided as Supplementary Data 2. Additional DFT results using explicit solvent molecules, calculated absorption coefficients, fits of the linear IR spectra, and 2D-IR spectra with parallel polarization combinations are available in the Supplementary Material file attached to this paper.

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

## Acknowledgements

We acknowledge funding by the Deutsche Forschungsgemeinschaft (DFG, German Research Foundation)—SFB1551—Project No. 464588647. G. Giubertoni acknowledges that this publication is part of the project (with Project Number VI.Veni.212.240) of the research program NWO Talent

Program Veni 2021, which is financed by the Dutch Research Council (NWO).

## Author contributions

F.M.: Conceptualization, Investigation, Formal Analysis, Writing—Original Draft Preparation, Writing—Review & Editing. P.G.A.: Formal Analysis, Writing—Review & Editing. M.B.: Conceptualization, Supervision, Writing—Review & Editing. G.G.: Conceptualization, Formal Analysis, Supervision, Writing—Review & Editing. J.H.: Conceptualization, Formal Analysis, Supervision, Writing—Review & Editing.

## Funding

## Competing interests

The authors declare no competing interests.
