## [Transparent Peer Review file · Communications Chemistry]

Stereoelectronic and Hydrogen-Bonding Effects on Hydroxyproline Conformation

Corresponding Author: Dr Johannes Hunger

Version 0:

Reviewer comments:

Reviewer #1

(Remarks to the Author)

This study presents differences in the infrared spectra of Boc-4S-Hyp-OMe and Boc-4R-Hyp-OMe dissolved in CHCl₃ to understand the molecular origin of collagen stability. Here, Boc-4S-Hyp-OMe and Boc-4R-Hyp-OMe are simplified models, and CHCl₃ was used as the solvent, providing a nonaqueous environment. The investigation methods employed here were conventional FTIR spectroscopy, 2D-IR spectroscopy, and DFT computations. The main conclusion from this study is that, in addition to the $n \rightarrow \pi^*$ interactions moderately stabilizing the trans conformations of both Boc-4S-Hyp-OMe and Boc-4R-Hyp-OMe, the H-bond interaction plays a critical role in governing hydroxyproline conformational equilibria and underscores its importance for collagen structure and assembly. Boc-4S-Hyp-OMe can form an intramolecular hydrogen bond, whereas Boc-4R-Hyp-OMe forms only intermolecular hydrogen bonds, leading to distinct aggregation behaviors in nonaqueous solution. The conclusions are reasonable based on the experimental and computational studies.

This reviewer considers the concentration-dependent FTIR spectra, shown in Figure 5, to provide the most compelling evidence, supporting the conclusions among the three categories of investigation results. While the FTIR spectra of Boc-4S-Hyp-OMe show little variation across different concentrations, those of Boc-4R-Hyp-OMe exhibit pronounced concentration-dependent changes. The concentration-dependent FTIR spectra of Boc-4R-Hyp-OMe in the C=O stretch region (Figure 5(b)) and O-H stretch region (Figure 5(d)) strongly support the conclusion. The peak at ~ 1675 cm⁻¹ is likely to originate from aggregates, formed by intermolecular H-bonding. Compared to the FTIR study, 2D-IR study is not likely to provide strong evidence for the conclusions. For example, the uncertainty in the anisotropy of the cross peaks is rather large. In addition, there is no estimation of the coupling between the two coupled vibrational modes, e. g., the peaks at 1755 and 1690 cm⁻¹ for Boc-4S-Hyp-OMe. Although this study used 2D-IR spectroscopy, a difficult and advanced experimental technique, the results only partially supported the conclusion. This work is considered publishable; however, this reviewer has several technical questions that could be answered to enhance its visibility. The questions are as follows:

1. In Figure 3(d), the anisotropy value of the 1755 cm⁻¹ peak was estimated to be approximately 0.3. However, the method of obtaining this value was not mentioned. To assess the anisotropy, it is necessary to compare 2D IR spectra collected with different pump-probe polarization configurations (parallel and perpendicular). In the manuscript, only 2D IR spectra collected with the parallel polarization configuration are shown. It is recommended to present 2D IR spectra in the parallel configuration.
2. The method used to estimate the absorbance differences (ΔOD) presented in Figures 3(d), 4(d), and 4(e) is unclear. These absorbance differences are pretty small, approximately on the order of 10^{-6} OD. The absolute values of these differences can be determined with the narrowband pump-broadband probe method used by Hamm, Lim, and Hochstrasser in 1998 (JPC B, 1998, 102(31), 6123-6138). Since all the 2D-IR spectra shown here were obtained through Fourier transform, the peak magnitudes are not directly measured as ΔOD (or ΔmOD). Surely, the magnitudes of the peaks are proportional to ΔOD . This part needs clarification.
3. The conclusions could be further strengthened if the coupling strength between the diagonal peaks is measured, although this may not be so feasible.
4. In Figure S1, an FTIR spectrum of 10 mM Boc-4R-Hyp-OMe in chloroform is shown. To more clearly visualize the increasing degrees of aggregation with increasing concentration, it is recommended to plot the normalized FTIR spectra of

the 100 mM and 10 mM samples on top of each other.

Reviewer #2

(Remarks to the Author)

This article presents a careful investigation into the molecular origins of collagen stability, specifically focusing on the conformational differences between (2S,4R)-hydroxyproline (4R-Hyp) and its diastereomer (2S,4S)-hydroxyproline (4S-Hyp). The study addresses a long-standing question in structural biology: why the minor stereochemical change from 4R to 4S-Hyp completely destabilizes the collagen triple helix. While previous literature has heavily emphasized stereoelectronic effects (the gauche effect and $n \rightarrow \pi^*$ interactions), this paper shifts the focus toward the critical role of hydrogen bonding. The authors have investigated the structures of 4R-Hyp and 4S-Hyp using linear IR and 2D IR spectroscopy coupled with DFT calculations.

The authors have demonstrated that 4S-Hyp is stabilized by a strong intramolecular hydrogen bond between the hydroxyl group and the ester carbonyl, which locks it into a conformation incompatible with the triple helix. 4R-Hyp lacks this intramolecular hydrogen bond and instead shows a high propensity for aggregation through intermolecular hydrogen bond, which is likely a key driver in the assembly of collagen fibrils. They have interpreted their studies to indicate that this intermolecular hydrogen bond exists between the amide group of a Boc-4R-Hyp-OMe molecule and the hydroxyl group of another molecule. They have concluded that hydrogen bond formation (not just stereoelectronic effects) in naturally occurring 4R-Hyp may be a key element for collagen's structural and assembly properties.

However, this entire conclusion on the mechanism of the collagen stability based on the study of only one amino acid residue Hyp and monitoring their self-assembly is difficult to believe. There were a tremendous number of studies in the literature on collagen and collagen models, especially by Raines and coworkers to substantiate the mechanism of stability of collagen and the role of 4R Hyp and why there cannot be 4S Hyp instead of 4R Hyp. It has been demonstrated that (ProFlpGly)₁₀ collagen model peptide forms stable collagen triple helix and this is much more stable than the collagen triple helix formed by (ProHypGly)₁₀ peptide model (Nature, 392, 1998, 666-667). In Flp, OH of Hyp is replaced by fluorine and there is no chance of intermolecular hydrogen bonding with F. The crystal structure of collagen rather shows interstrand N-H(Gly)...O=C(Xaa) hydrogen bonds which held the strands together while a series of $n-\pi^*$ interactions drive the stability of very long individual strand. Does the crystal structure of collagen show intermolecular hydrogen bond between the amide group of a 4R-Hyp of one strand and the hydroxyl group of 4R-Hyp of another strand?

Hence, this manuscript cannot be accepted for publication in Communications Chemistry.

Version 1:

Reviewer comments:

Reviewer #1

(Remarks to the Author)

The authors have revised the manuscript in response to the reviewer's comments. In the reviewer's opinion, the revised version represents a significant improvement over the previous submission and is now suitable for publication.

Reviewer #2

(Remarks to the Author)

The authors now addressed most of the questions/comments asked on the manuscript and have revised the manuscript accordingly. Hence, the manuscript can be accepted for publication in communications Chemistry.

Reviewers' comments:

Reviewer #1 (Remarks to the Author):

This study presents differences in the infrared spectra of Boc-4S-Hyp-OMe and Boc-4R-Hyp-OMe dissolved in CHCl₃ to understand the molecular origin of collagen stability. Here, Boc-4S-Hyp-OMe and Boc-4R-Hyp-OMe are simplified models, and CHCl₃ was used as the solvent, providing a nonaqueous environment. The investigation methods employed here were conventional FTIR spectroscopy, 2D-IR spectroscopy, and DFT computations. The main conclusion from this study is that, in addition to the $n \rightarrow \pi^*$ interactions moderately stabilizing the trans conformations of both Boc-4S-Hyp-OMe and Boc-4R-Hyp-OMe, the H-bond interaction plays a critical role in governing hydroxyproline conformational equilibria and underscores its importance for collagen structure and assembly. Boc-4S-Hyp-OMe can form an intramolecular hydrogen bond, whereas Boc-4R-Hyp-OMe forms only intermolecular hydrogen bonds, leading to distinct aggregation behaviors in nonaqueous solution. The conclusions are reasonable based on the experimental and computational studies.

This reviewer considers the concentration-dependent FTIR spectra, shown in Figure 5, to provide the most compelling evidence, supporting the conclusions among the three categories of investigation results. While the FTIR spectra of Boc-4S-Hyp-OMe show little variation across different concentrations, those of Boc-4R-Hyp-OMe exhibit pronounced concentration-dependent changes. The concentration-dependent FTIR spectra of Boc-4R-Hyp-OMe in the C=O stretch region (Figure 5(b)) and O-H stretch region (Figure 5(d)) strongly support the conclusion. The peak at $\bar{1}675$ cm⁻¹ is likely to originate from aggregates, formed by intermolecular H-bonding. Compared to the FTIR study, 2D-IR study is not likely to provide strong evidence for the conclusions. For example, the uncertainty in the anisotropy of the cross peaks is rather large. In addition, there is no estimation of the coupling between the two coupled vibrational modes, e. g., the peaks at 1755 and 1690 cm⁻¹ for Boc-4S-Hyp-OMe. Although this study used 2D-IR spectroscopy, a difficult and advanced experimental technique, the results only partially supported the conclusion. This work is considered publishable; however, this reviewer has several technical questions that could be answered to enhance its visibility. The questions are as follows:

1. In Figure 3(d), the anisotropy value of the 1755 cm⁻¹ peak was estimated to be approximately 0.3. However, the method of obtaining this value was not mentioned. To assess the anisotropy, it is necessary to compare 2D IR spectra collected with different pump-probe polarization configurations (parallel and perpendicular). In the manuscript, only 2D IR spectra collected with the parallel polarization configuration are shown. It is recommended to present 2D IR spectra in the parallel configuration.

Authors' reply:

Thank you for addressing this point. We fully agree that in our original submission, the methodological description regarding the anisotropy was too brief. We have revised the description and also included 2D-IR spectra recorded with parallel polarization combinations.

Authors' action:

We changed the following parts in the main text on Page 7:

"We then calculated the anisotropy of the cross-peak signals, defined as $R = \frac{I_{\text{par}} - I_{\text{perp}}}{I_{\text{par}} + 2I_{\text{perp}}}$, where I_{par} and I_{perp} are the 2D-IR signal intensities recorded with the probe pulses polarized parallel and perpendicular relative to the excitation pulses, respectively (parallel 2D-IR spectra are shown in the Supporting Information). In the case of cross-peaks, the anisotropy can be used to estimate the angle, θ , between the transition dipoles of the two coupled vibrations as $R = (3\cos^2\theta - 1)/5$ [42-44]. To obtain R for the cross-peak observed when exciting at 1690 cm^{-1} and probing at 1755 cm^{-1} , ...". Further, we included parallel 2D-IR spectra in the Supporting Information.

Fig. R1 2D-IR spectra of Boc-4S-Hyp-OMe in chloroform at 200 mM concentration in parallel polarization at a waiting time of 0.3 ps.

Fig. R2 2D-IR spectra of Boc-4R-Hyp-OMe in chloroform at 200 mM concentration in parallel polarization at a waiting time of 0.3 ps.

2. The method used to estimate the absorbance differences (ΔOD) presented in Figures 3(d), 4(d), and 4(e) is unclear. These absorbance differences are pretty small, approximately on the order of 10^{-6} OD. The absolute values of these differences can be determined with the narrowband pump-broadband probe method used by Hamm, Lim, and Hochstrasser in 1998 (JPC B, 1998, 102(31), 6123-6138). Since all the 2D-IR spectra shown here were obtained through Fourier transform, the peak magnitudes are not directly measured as ΔOD (or ΔmOD). Surely, the magnitudes of the peaks are proportional to ΔOD . This part needs clarification.

Authors' reply:

Thank you for pointing this out. Indeed, the axis labels (ΔmOD) in Figures 3(d), 4(d), and 4(e) of our original submission were incorrect as they were obtained from Fourier-transformed free induction decays. We apologize for this mistake, which we have corrected in the revised manuscript.

Authors' action:

The labels of Figures 3(d), 4(d), and 4(e) are changed from ΔmOD to Intensity (arb. unit). We also included the following sentence in the method section: "The resulting intensity of the 2D-IR signals is proportional to the difference between the probe absorption with and without the pump pulse."

Fig. R3 (a) Linear IR spectrum (center panel) and its second derivative (bottom panel) of a 200 mM solution of Boc-4S-Hyp-Ome in chloroform in the amide and ester stretching region. The top panel displays the conformations of the amide and ester groups, which are represented by bars of the same colors in the center panel. (b) Structures of Boc-4S-Hyp-Ome obtained from conformational search and geometry optimization by DFT calculations. In the bottom table, the dihedral angles of the amide group ω and ester group ψ , Gibbs free energy difference from conformer (A) ΔG , and the anisotropy parameter R_{DFT} calculated from the structures are displayed. Classification of conformers by the dihedral angles of the amide group ω (trans or cis) is shown in parentheses next to the dihedral angle values. (c) 2D-IR spectrum of the 200 mM Boc-4S-Hyp-Ome solution in chloroform in perpendicular polarization at a waiting time of 0.3 ps. In the gray boxes, intensities are magnified 15-fold to emphasize cross-peaks. The dotted line indicates the pump frequency corresponding to the amide stretching vibration of the trans conformer. (d) Pump slice at 1690 cm^{-1} (top) and the corresponding anisotropy spectrum (bottom) as indicated by the dotted line in panel (c). The corresponding conformer is indicated by an arrow.

Fig. R4 (a) Linear IR spectrum (center panel) and the second derivative (bottom panel) of 200 mM Boc-4R-Hyp-OMe in chloroform in the amide and ester stretching region. The top panel shows the conformations of the amide and ester groups underlying the bar plots of the same color in the center panel. (b) The structures of Boc-4R-Hyp-OMe obtained from conformational search and geometry optimization by DFT calculations. In the bottom table, the dihedral angles of the amide group ω and ester group ψ , Gibbs free energy difference from conformer (E) ΔG , and the anisotropy parameter R_{DFT} calculated from the structures are displayed. Classification of conformers by the dihedral angles of the amide group ω (trans or cis) is shown in parentheses next to the dihedral angle values. (c) 2D-IR spectrum of 200 mM Boc-4R-Hyp-OMe in chloroform in perpendicular polarization at a waiting time of 0.3 ps. In the gray boxes, intensities are magnified 15-fold to emphasize cross-peaks. The dotted lines indicate the pump frequency corresponding to the amide stretching vibration of the trans (orange) and cis (green) conformers. Pump slices (top) and anisotropy spectra (bottom) at (d) 1690 cm^{-1} and (e) 1705 cm^{-1} , as indicated by the dotted lines in panel (c). The corresponding conformer for the peak is indicated with an arrow.

3. The conclusions could be further strengthened if the coupling strength between the diagonal peaks is measured, although this may not be so feasible.

Authors' reply:

Thank you for the excellent suggestion. We fully agree that our results would be strengthened by such analysis; Following the Reviewer's suggestion, we attempted to extract the coupling strength from the 2D-IR spectra based on peak intensities and anharmonicities. However, as the Reviewer anticipated, this was not feasible for our system due to overlapping spectral contributions from different conformations in the diagonal peaks and insufficient frequency separation between the cross-peak signals and the tails of the diagonal peaks.

Authors' action

We included the following description in the main text on Page 7: In addition to the contribution of the tail of the diagonal peaks to the cross-peaks, also overlapping spectral contributions in the diagonal peaks prevent a more quantitative evaluation of the coupling strengths between amide and ester vibrational modes in our system.

4. In Figure S1, an FTIR spectrum of 10 mM Boc-4R-Hyp-OMe in chloroform is shown. To more clearly visualize the increasing degrees of aggregation with increasing concentration, it is recommended to plot the normalized FTIR spectra of the 100 mM and 10 mM samples on top of each other.

Authors' reply:

Thank you for this insightful suggestion. We plotted the concentration-dependent spectra of Boc-4S- and Boc-4R-Hyp-OMe using 10 and 100 mM solutions for clearer visualization. The spectra at amide frequencies are normalized to the total peak area within the displayed region. Because the transition dipoles of hydrogen-bonded and free OH groups differ significantly, we scaled the OH spectra solely by the concentration ratio.

Authors' action:

We revised Figure 5 by plotting 10 and 100 mM concentrations instead of 100, 200, and 400 mM.

Fig. R5 The concentration-dependent (10 and 100 mM) linear IR spectra of Boc-4S- and Boc-4R-Hyp-OMe in chloroform solution in the amide stretching ((a) and (b)) and OH stretching ((c) and (d)) region. The IR spectra were normalized to the total area of the peaks in the displayed region for panels (a) and (b). Due to the largely differing transition dipole moments of hydrogen-bonded and non-hydrogen-bonded OH stretching modes, the spectra in panels (c) and (d) were scaled to account for the different concentrations.

Reviewer #2 (Remarks to the Author):

This article presents a careful investigation into the molecular origins of collagen stability, specifically focusing on the conformational differences between (2S,4R)-hydroxyproline (4R-Hyp) and its diastereomer (2S,4S)-hydroxyproline (4S-Hyp). The study addresses a long-standing question in structural biology: why the minor stereochemical change from 4R to 4S-Hyp completely destabilizes the collagen triple helix. While previous literature has heavily emphasized stereoelectronic effects (the gauche effect and $n \rightarrow \pi^*$ interactions), this paper shifts the focus toward the critical role of hydrogen bonding. The authors have investigated the structures of 4R-Hyp and 4S-Hyp using linear IR and 2D IR spectroscopy coupled with DFT calculations.

The authors have demonstrated that 4S-Hyp is stabilized by a strong intramolecular hydrogen bond between the hydroxyl group and the ester carbonyl, which locks it into a conformation incompatible with the triple helix. 4R-Hyp lacks this intramolecular hydrogen bond and instead shows a high propensity for aggregation through intermolecular hydrogen bond, which is likely a key driver in the assembly of collagen fibrils. They have interpreted their studies to indicate that this intermolecular hydrogen bond exists between the amide group of a Boc-4R-Hyp-OMe molecule and the hydroxyl group of another molecule. They have concluded that hydrogen bond formation (not just stereoelectronic effects) in naturally occurring 4R-Hyp may be a key element for collagen's structural and assembly properties.

However, this entire conclusion on the mechanism of the collagen stability based on the study of only one amino acid residue Hyp and monitoring their self-assembly is difficult to believe. There were a tremendous number of studies in the literature on collagen and collagen models, especially by Raines and coworkers to substantiate the mechanism of stability of collagen and the role of 4R HyP and why there cannot be 4S HyP instead of 4R HyP. It has been demonstrated that (ProFlpGly)₁₀ collagen model peptide forms stable collagen triple helix and this is much more stable than the collagen triple helix formed by (ProHypGly)₁₀ peptide model (Nature, 392, 1998, 666-667). In Flp, OH of Hyp is replaced by fluorine and there is no chance of intermolecular hydrogen bonding with F. The crystal structure of collagen rather shows interstrand N-H(Gly)...O=C(Xaa) hydrogen bonds which held the strands together while a series of $n\text{-}\pi^*$ interactions drive the stability of very long individual strand. Does the crystal structure of collagen show intermolecular hydrogen bond between the amide group of a 4R-Hyp of one strand and the hydroxyl group of 4R-Hyp of another strand?

Hence, this manuscript cannot be accepted for publication in Communications Chemistry.

Authors' reply:

We thank the Reviewer for their critical feedback. The Reviewer criticizes the limited relevance of the intermolecular aggregation observed here for small model peptides to interpeptide interactions in full-length collagen and, consequently, to the stability and self-assembly of collagen. Indeed, while our study was inspired by the role of 4R-Hyp in collagen stability, we emphasize that our investigation focuses on conformational equilibria and interactions at the level of single amino acids. At this scale, our work challenges previous findings on the effect of stereochemistry on conformation; specifically, we demonstrate that conformational analysis is strongly influenced by intermolecular interactions that were previously overlooked. As such, our results provide critical insights into mechanisms that stabilize amino-acid conformations at the monomeric level.

Further, we show that differing stereochemistry dictates distinct aggregation behaviors. Yet, as correctly pointed out by the Reviewer, this different aggregation behavior cannot be directly transferred to real collagen fibrils. Nevertheless, our results show that the different aggregation tendencies are due to different levels of exposure of the OH group of 4R-Hyp and 4S-Hyp to the environment. This exposure of the OH group of 4R-Hyp is almost certainly similar at the single-amino-acid level and in full collagen. In fact, the crystal structure of collagen reported by Bella et al. (1994), in which the 4R-Hyp hydroxyl group was identified as a keystone for hydration, forming robust intermolecular H-bonds with surrounding water molecules, is consistent with this notion.

Authors' action:

To prevent misinterpretation of our results regarding their transferability to collagen, we emphasize that the observed aggregation behavior does not concern collagen stability and does not imply that intermolecular Hyp-Hyp hydrogen bonds are formed during collagen assembly or within the helix in the revised manuscript. Specifically, we have implemented the following major revisions:

- (i) We have carefully revised any sentences that could indicate transferability of the single-residue level to full collagen throughout the manuscript.
- (ii) We have moved all hypothesized implications of the present findings (e.g. the different exposure of the OH group to the environment) to a paragraph at the end of the discussion, where we discuss the potential implications but also the limitations of the single-residue study.
- (iii) We have elaborated on the previous work on fluorinated proline, which indeed can isolate stereoelectronic effects.

Page 1, Abstract:

Conversely, our results show that hydrogen bonding is pivotal: the conformation of Boc-4S-Hyp-OMe is stabilized by an intramolecular hydrogen bond, whereas Boc-4R-Hyp-OMe primarily forms intermolecular hydrogen bonds, leading to a greater intermolecular affinity. These findings highlight hydrogen bonding as a key determinant of hydroxyproline conformation at the single-residue level. ~~These insights reveal~~

~~the pivotal role of hydrogen bonding in shaping hydroxyproline's conformation and interactions—features essential to the stability and biological performance of collagen.~~

Page 3, Introduction:

The significance of such stereoelectronic effects is demonstrated in earlier works on collagen mimetic peptides containing 4R-fluoroproline (Flp), for which fluorination enforces stereoelectronic effects without incorporating a strong hydrogen bond donor. In these studies, fluorination has been shown markedly stabilize the triple helical structure (Holmgren et al., *Nature*, 392, 6677, 666-667(1998) and Holmgren et al., *Chem. Biol.*, 6, 2, 63-70(1999)).

Page 4, Introduction:

To elucidate their role in the stability of collagen, the complexity arising from conformational flexibility and solute-solvent interactions has to be disentangled also at the single-residue level.

Page 4, Introduction:

Conversely, Boc-4R-Hyp-OMe can only form intermolecular hydrogen bonds; this is reflected in its preference for forming intermolecular hydrogen bonds with other molecules, leading to self-aggregation for the present samples.

Page 12, Discussion:

As such, we find that the intramolecular hydrogen bond not only determines the ring geometry but also the orientation of the ester group, ~~which is critical for the conformation of larger collagen peptides.~~

Page 12, Discussion:

~~Our results show that the intramolecular hydrogen bond of 4S-Hyp competes with this type of intermolecular interaction. Such a competition of intra- and intermolecular hydrogen bonds in 4S-Hyp is in line with the instability of the collagen-mimicking peptides containing 4S-Hyp.~~

Page 12, Discussion

Finally, although the single-residue level of the present work cannot be directly related to larger systems such as full collagen fibrils, our results may have potential implications for larger systems. Our results indicate that Boc-4R-Hyp-OMe tends to form intermolecular interactions resulting in enhanced aggregation as compared to Boc-4S-Hyp-OMe. In full collagen fibrils, the conformational constraints due to other residues and proteins render such direct interactions between collagen residues very unlikely.

Nevertheless, the self-aggregation observed here reflects a pronounced tendency of Boc-4R-Hyp-OMe to expose its hydroxyl group to the environment, resulting in intermolecular interactions rather than intramolecular interactions. This observation is consistent with the notion that naturally occurring 4R-Hyp in collagen functions as a keystone within the hydration network and as a potential recognition site during assembly (Bella et al., *Science*, 266, 5182,75–81 (1994) and Khare et al., *Proc. Nat. Acad. Sci.*,119,40,e2209524119 (2022)), both of which require exposure of 4R-Hyp's OH group to the environment.

Page 12, Conclusion

Thus, our findings not only refine our understanding of the stabilizing mechanisms of hydroxyproline conformation but highlight the importance of hydrogen bonding in determining its conformational equilibria and molecular interaction tendencies at the level of a single amino acid residue.

~~Although more studies are required to establish the role of hydrogen bonding of hydroxyproline to water and/or other proteins for collagen stability, our results suggest that hydrogen bond formation (not just stereoelectronic effects) in naturally occurring 4R-Hyp may be a key element for collagen's structural and assembly properties.~~